# The forests of the midwestern United States at Euro-American settlement: Spatial and physical structure based on contemporaneous survey data

**Christopher J. Paciorek**[1]*, **Charles V. Cogbill**[2], **Jody A. Peters**[3], **John W. Williams**[4,5], **David J. Mladenoff**[6], **Andria Dawson**[7], **Jason S. McLachlan**[3]

**1** Department of Statistics, University of California, Berkeley, California, United States of America, **2** Harvard Forest, Harvard University, Petersham, Massachusetts, United States of America, **3** Department of Biological Sciences, University of Notre Dame, Notre Dame, Indiana, United States of America, **4** Department of Geography, University of Wisconsin, Madison, Wisconsin, United States of America, **5** Center for Climatic Research, University of Wisconsin, Madison, Wisconsin, United States of America, **6** Department of Forest and Wildlife Ecology, University of Wisconsin, Madison, Wisconsin, United States of America, **7** Department of General Education, Mount Royal University, Calgary, Alberta, Canada

* paciorek@stat.berkeley.edu

**Data Availability Statement:** The following data products are available via the LTER Network Data Portal. • Raw gridded stem density, aboveground

## Abstract

We present gridded 8 km-resolution data products of the estimated stem density, basal area, and biomass of tree taxa at Euro-American settlement of the midwestern United States during the middle to late 19th century for the states of Minnesota, Wisconsin, Michigan, Illinois, and Indiana. The data come from settlement-era Public Land Survey (PLS) data (ca. 0.8-km resolution) of trees recorded by land surveyors. The surveyor notes have been transcribed, cleaned, and processed to estimate stem density, basal area, and biomass at individual points. The point-level data are aggregated within 8 km grid cells and smoothed using a generalized additive statistical model that accounts for zero-inflated continuous data and provides approximate Bayesian uncertainty estimates. The statistical modeling smooths out sharp spatial features (likely arising from statistical noise) within areas smaller than about 200 km$^2$. Based on this modeling, presettlement Midwestern landscapes supported multiple dominant species, vegetation types, forest types, and ecological formations. The prairies, oak savannas, and forests each had distinctive structures and spatial distributions across the domain. Forest structure varied from savanna (averaging 27 Mg/ha biomass) to northern hardwood (104 Mg/ha) and mesic southern forests (211 Mg/ha). The presettlement forests were neither unbroken and massively-statured nor dominated by young forests constantly structured by broad-scale disturbances such as fire, drought, insect outbreaks, or hurricanes. Most forests were structurally between modern second growth and old growth. We expect the data product to be useful as a baseline for investigating how forest ecosystems have changed in response to the last several centuries of climate change and intensive Euro-American land use and as a calibration dataset for paleoecological proxy-based reconstructions of forest composition and structure for earlier time periods.

biomass and basal area (DOI: 10.6073/pasta/801601af769fa5acade1ef07f6892bdd). • Gridded statistically-smoothed stem density (DOI: 10.6073/pasta/1b2632d48fc79b370740a7c20a70b4b0). • Gridded statistically-smoothed aboveground biomass (DOI: 10.6073/pasta/b246e05afb25dbe06b3006c5d18a4a2b). • Gridded statistically-smoothed basal area (DOI: 10.6073/pasta/c3ae2363e4ae2e0f42a7c02b6f12b50a). The point-level raw data are available as well: • Indiana (DOI:10.6073/pasta/c3e2404f5b34204b5871a743ebce3c51) • Illinois (DOI: 10.6073/pasta/b8fdabd7cfba3a2b3d55fe4c1dc5383f) • Michigan ○ Southeastern (DOI: 10.6073/pasta/409ec6dfb218b6a3e98022916d2b4438) ○ Southern (DOI: 10.6073/pasta/8d033c1cfadca42bf060f9f38940c81e) ○ Northern (DOI: 10.6073/pasta/3760eec82562e0a8b7cd493c0a3e3ef4) • Wisconsin (DOI: 10.6073/pasta/c3e680e51026e74a103663ffa16cb95d) • Minnesota (DOI: 10.6073/pasta/f55f6b7f4060a9b4f07374e7db8443cd) The project GitHub repository (https://github.com/PalEON-Project/PLS_products) provides code for processing the point-level data and producing the data products above in the subdirectory named 'R'. In the subdirectory 'data/conversions', we provide: • Our translation tables for translating surveyor taxon abbreviations to modern common names, including aggregation for the raw gridded values and statistical modeling done in this work, • Correction factors for the subregions of the domain for estimating point-level tree density, and, • Our assignments of allometric relationships for the PalEON taxa, based on [22].

**Funding:** The authors acknowledge support from the National Science Foundation (nsf.gov) from grants 1241874 (CJP, CVC, JAP, AD, JSM) and 1241868 (CVC, JWW, AD). The funders had no role in study design, data collection and analysis, decision to publish, or preparation of the manuscript.

**Competing interests:** The authors have declared that no competing interests exist.

The data products (including raw and smoothed estimates at the 8-km scale) are available at the LTER Network Data Portal as version 1.0.

## Introduction

Terrestrial vegetation in midwestern North America changed drastically at the time of Euro-American settlement [1,2]. Before settlement, the midwestern United States was the location of a major ecological transition between the grasslands of the Great Plains and the forests of eastern and northern North America [3–5]. These grasslands have now mostly been replaced by agriculture or pastoral land use, except in areas of prairie conservation or restoration. Forested areas were also heavily affected by clearance for agriculture and logging during and after settlement [2,6,7]. Historical vegetation surveys from this time period, collected during the time of land surveys and allotment, provide critical context for understanding terrestrial ecosystems, the carbon cycle, and vegetation-atmosphere feedbacks [2,8,9]. These datasets allow researchers to define 'baseline' conditions for purposes of conservation planning [10], to understand ecosystem processes at decadal and centennial scales [11], to track how vegetation changes with changing climate [12,13], to understand changes in ecosystems after widespread land use change [2,14], and to calibrate paleoecological data [15,16]. The presettlement composition and vegetation types for the Midwestern states from Ohio to Minnesota have previously been comprehensively mapped, for example [17–22]. Here we quantify the structure (distribution and size) of forests, woodlands, and grasslands across the Midwest using statistical methods that provide the first statistically robust estimates of stem density, basal area, and biomass preceding 19th century land clearance.

Euro-American settlement and subsequent land use change occurred over many decades across the Midwest. During that time, surveys demarcated land for land tenure and use, usually involving recording and marking specific ("witness") trees adjacent to survey corners [8,9,17]. These data provide information on tree taxon identifications, sizes, and distributions that can be mapped and used quantitatively to represent forest composition, and, sometimes, structure at the period of settlement. In the northeastern United States, early surveys provide only data at the township level [23,24], which cannot be used to estimate stem density, basal area, and biomass, but which we have used to estimate composition [22]. Later surveys after the establishment of the U.S. Public Land Survey System (PLS) by the General Land Office (GLO) provide point-level (i.e., corner-level) data along a regular grid, every one-half mile (~805 m) spacing, for Ohio and westward during the period 1785 to 1907 [7,25–27]. At each point surveyors identified two to four trees and recorded the common name, diameter at breast height (dbh), and distance and bearing from the point to the trees. Using plotless density estimation techniques [28], we can estimate stem density, basal area, and biomass at each point from these data. These point-level data are quite noisy, but they can be aggregated to coarser spatial resolution to more robustly estimate spatial patterns of vegetation [7]. At the 8 km grid resolution, the estimates are still noisy, and there are some spatial gaps in the available data, so in this work we employ a spatial statistical model to smooth over the noise and impute values for missing grid cells. The result is a statistical data product that provides statistical estimates of biomass and density with quantitative estimates of uncertainty.

This work builds upon and extends beyond prior work described in [7,22]. In [22], we used these data to estimate forest composition. In this work, we estimate stem density, basal area, and biomass using an extended dataset. In [7], we estimated stem density, basal area, and

biomass for a smaller domain using an earlier version of the point-level data. This earlier paper focused on comparing settlement-era and modern vegetation composition with an emphasis on identifying forest types that had largely disappeared after land use and the emergence of novel forests. Here, we build upon [7] by using a new spatial statistical model to smooth over the noisy grid-level estimates; extending the domain to roughly double the geographic coverage by including Illinois, Indiana, and southern Michigan; using updated allometric scaling factors from [29]; and applying additional and consistent data cleaning steps across the region. We use the updated estimates to analyze the biogeographic patterns in vegetation across the domain.

In Methods, we first describe the procedures used to obtain and clean the PLS survey data at the survey points. Second, we describe the processing that homogenizes the data across the states of interest. Third we describe the statistical methodology used to estimate stem density, basal area, and biomass; first at the individual survey points, and then on an 8 km by 8 km grid, stratified by taxon and in total. Finally, we describe the various data products we have produced and archived. In Results we first present basic summaries of stem density, basal area, and biomass as maps and regional averages, assessing the spatial variation across the region. We then compare the presettlement to contemporary estimates. Finally, in Discussion we summarize our understanding of the presettlement landscape in light of the new data products and discuss the uncertainties estimated by the statistical model and the limitations of the model.

This work provides the first statistically-rigorous estimates of forest structure at reasonably high spatial resolution over a large spatial area. The statistical approach to estimating the spatial patterns is new and effective, and its estimates allow us to report on the structure and spatial variability of ecosystems at the time of settlement. These estimates provide a baseline and a calibration dataset for researchers interested in assessing the transformation of the spatial patterns and structure of ecosystems by intensive Euro-American land use.

## Methods

### PLS data collection and cleaning

The PLS was developed to enable the division and sale of US federal lands from Ohio westward. The survey created a 1 mile$^2$ (2.59 km$^2$) grid (sections) on the landscape. At each half-mile (quarter-section) and mile (section) survey point, a post was set, or a tree was blazed as the official location marker. PLS surveyors then recorded tree stem diameters, measured distances and bearings of the two to four trees adjacent to the survey point, and identified tree taxa using common (and often regionally idiosyncratic) names. In the Midwest, PLS data thus represent measurements by hundreds of surveyors from 1786 until 1907, with changing sets of instructions over time [30,31]. Survey procedures varied widely in Ohio, and distance, diameter, and bearing information are not systematically available, so Ohio is not included in this work. The work presented here builds upon prior digitization and classification of PLS data for Wisconsin, Minnesota, and Michigan [7], with extensive additional cleaning and correction of the Michigan data and extensive additional digitization of Illinois and Indiana by the authors (Fig 1). Digitization of PLS data in Minnesota, Wisconsin and Michigan's Upper Peninsula and northern Lower Peninsula is essentially complete, with PLS data for nearly all 8 km grid cells. Data for the southern portion of Michigan's Lower Peninsula include the section points, but the quarter-section points have not been digitized yet, except for 27 townships in southeastern Michigan, which are complete. Data in Illinois and Indiana represent a sample of the full set of grid cells, with survey record transcription ongoing at the University of Notre Dame (Fig 1).

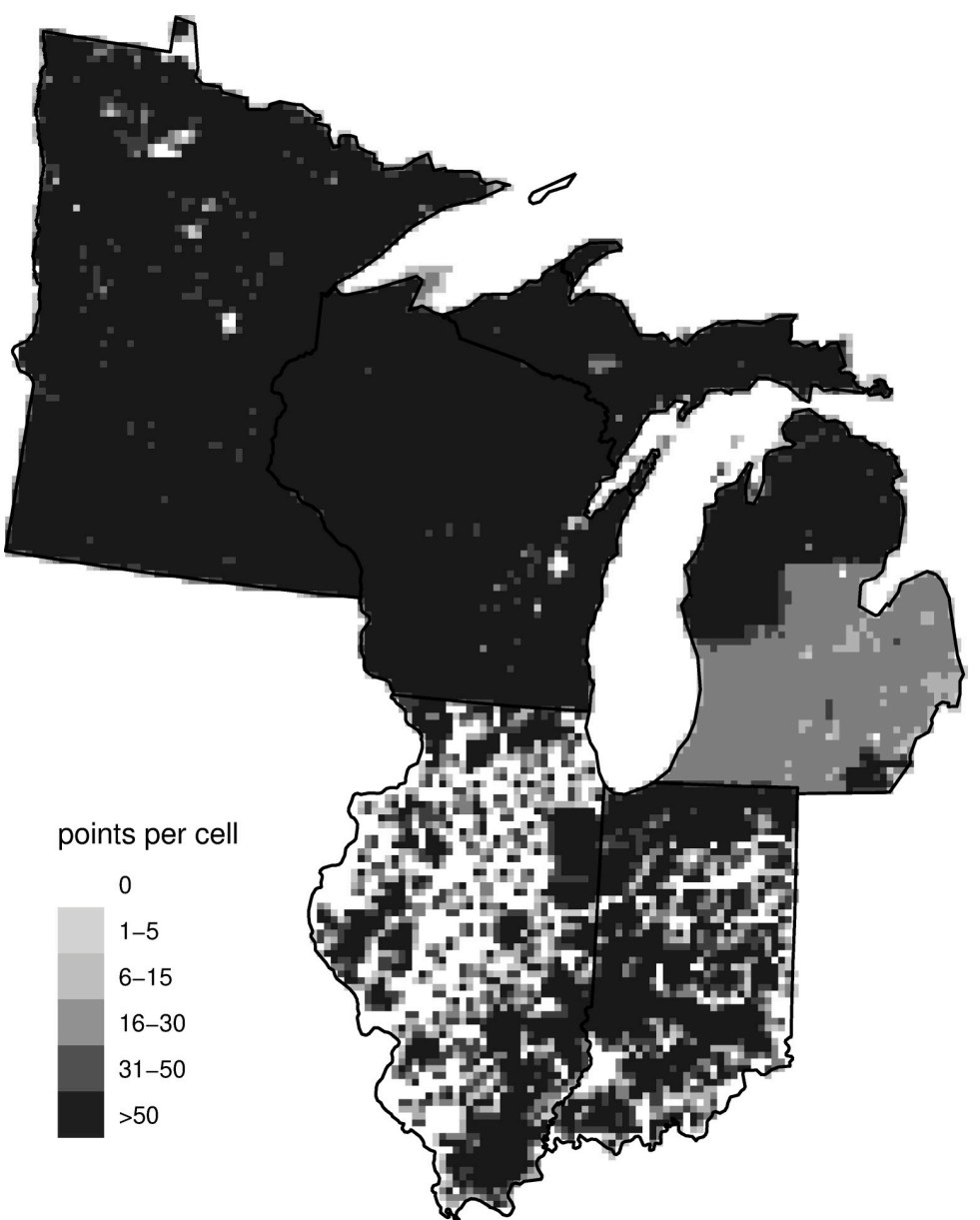

**Fig 1. Number of PLS points per 8-km grid cell.** Lighter grey in southern Michigan is caused by lack of quarter-section points. Illinois and Indiana digitization is ongoing.

As discussed in [22], the surveys in our domain occurred over a period of more than 100 years (starting in 1799 in Indiana and ending in 1907 in Minnesota) as settlers from the United States and Europe moved into what is now the midwestern United States. Our estimates are for the period of settlement represented by the survey data and therefore are time-transgressive. They do not represent any single point in time across the domain, but rather the state of the landscape at the time just prior to widespread logging and land clearance [23,32]. These datasets include the effects of Native American land use and early Euro-American settlement activities [33], but it is likely that the imprint of this earlier land use is locally concentrated rather than spatially extensive [34].

We used expert judgment (co-author Cogbill) and prior work to determine the current common names of surveyor-recorded vernacular terms and abbreviations of settlement-era common names. We then aggregated into taxonomic groups that are primarily at the genus level but include some monospecific genera. We use the following 20 taxa plus an "other hardwood" category: Ash (*Fraxinus* spp.), Basswood (*Tilia americana*), Beech (*Fagus grandifolia*), Birch (*Betula* spp.), Black gum/sweet gum (*Nyssa sylvatica* and *Liquidambar styraciflua*), Cedar/juniper (*Juniperus virginiana* and *Thuja occidentalis*), Cherry (*Prunus* spp.), Dogwood (*Cornus* spp.), Elm (*Ulmus* spp.), Fir (*Abies balsamea*), Hemlock (*Tsuga canadensis*), Hickory (*Carya* spp.), Ironwood (*Carpinus caroliniana* and *Ostrya virginiana*), Maple (*Acer* spp.), Oak (*Quercus* spp.), Pine (*Pinus* spp.), Poplar/tulip poplar (*Populus* spp. and *Liriodendron tulipifera*), Spruce (*Picea* spp.), Tamarack (*Larix laricina*), and Walnut (*Juglans* spp.). Note that because of several cases of ambiguity in the common tree names used by surveyors (black gum/sweet gum, ironwood, poplar/tulip poplar, cedar/juniper), a group can represent trees from different genera or even families.

In S1 Appendix, we describe the specific data-cleaning steps we applied to each sub-dataset as well as a variety of steps to standardize the dataset across states and minimize the potential effects of surveyor bias upon estimates of vegetation. Note that the division between northern and southern Michigan is caused by obtaining the data from different sources and can be seen in the differential data densities in Fig 1.

## Estimation of point-level quantities

**Point-level density.** We estimated stem density at each point with a Morisita plotless density estimator that uses the measured distances from each survey point to the nearest trees at the point location [28]. The standardized approach for the Morisita method is well-validated [28]. However, over time the survey design used by PLS surveyors changed as protocols were updated, which affects how we estimate density from the information at each point. S2 Appendix summarizes the changes in the information recorded and how we developed and applied spatially-varying correction factors to the Morisita estimator [7] to account for these changes when estimating stem density at a point. Distances from the tree to the survey point were taken to be the distance from the survey notes plus one-half the diameter of the tree.

In S1 Appendix, we detail the steps taken to either exclude points or adjust the density estimates at points where direct estimation of density was impossible or posed a risk of bias. After all removals we estimated stem density at 66,648 Illinois points, 67,072 Indiana points, 113,801 Michigan points, 226,047 Minnesota points, and 159,058 Wisconsin points (Fig 1).

We limited estimates of density to trees greater than or equal to 8 inches (20.32 cm) dbh because that is approximately the size below which surveyors tended to avoid sampling small trees. However, at many points smaller trees were reported by surveyors. Smaller trees reported by surveyors were a censored sample to some degree. We included all trees that were surveyed in our initial density estimate (including those with missing diameters), giving a raw stem density estimate whose meaning (in terms of the implicit diameter threshold) varies spatially based on how surveyors selected for tree size in a given area. We then used a spatially-varying correction factor [7] to scale the raw density estimates to a corrected stem density estimate for trees greater than or equal to 8 inches dbh.

**Estimation of individual tree biomass.** We use the allometric scalings from [29] to estimate aboveground biomass (AGB) from dbh. The assignment of allometric coefficients (for simple linear regressions of log biomass (kg) on log dbh (cm)) to taxa is provided in our GitHub repository (https://github.com/PalEON-Project/PLS_products). Note that some of the 21 taxa use the same allometric equations.

Our original goal was to make use of the full set of allometric information in [29,35] to incorporate uncertainty in scaling dbh to tree biomass, using the Bayesian statistical methods provided in the PEcAn software [36] allometry module. However, even at the taxonomic aggregation inherent in our 21 taxa, there are often few allometries available for a given taxonomic group, and, in many cases, the allometries come from locations outside of our midwestern US spatial domain. Furthermore, although there are more allometries for stem biomass (component 6 in the nomenclature of [35]; note that this excludes branches) than for aboveground (component 2) or total biomass (component 1), most research focuses on aboveground biomass rather than stem biomass. As a result, it was not feasible to robustly estimate the aboveground biomass allometries with uncertainty and therefore we have omitted incorporating allometric uncertainty.

**Estimation of point-level biomass and basal area.** Here we describe how we calculate biomass at each PLS point. Calculations for basal area follow an analogous process.

In the usual case of having two trees, we calculated the point-level biomass as one-half the stem density multiplied by the estimated biomass of each tree. When the two trees were from different taxa, this calculation produces point-level biomass for two taxa that were added to estimate total biomass. When both trees are from the same taxon, this is equivalent to averaging the tree-level biomass for the two trees and multiplying by stem density.

For simplicity we excluded all 3,221 points with any tree-level missing biomass values (i.e., missing diameters), although we note that it is possible to estimate (1) total biomass based on having one of two trees with available biomass and (2) taxon-level biomass from the available tree. Since one-tree points with missing biomass cannot be used for estimation, excluding two tree points with missing biomass data treats one- and two-tree points similarly, with the goal of limiting bias at the grid cell level.

To estimate biomass, we used the original density (i.e., without using the correction factors that account for size-biased sampling to estimate density of trees greater than or equal to 8 inches dbh) combined with biomass estimates for all individual trees (including those less than 8 inches dbh) to produce an unbiased biomass estimate without an explicit size threshold. We recognize that the different surveyor behavior regarding the diameter threshold introduces some imprecision, but the effect should be small given the limited contribution of smaller trees to total biomass. In contrast, for density estimation it is critical to define a diameter threshold in order to have a meaningful quantity.

## Statistical modeling at the grid scale

**Grid-level estimation.** Before statistically modeling at the 8-km grid scale, we aggregated the point-level data to the 8-km grid by averaging over point-level stem density, basal area, and biomass values for all points in a grid cell. In addition, for our statistical modeling to be able to account for the high abundance of points with either no trees or (for taxon-specific analyses) no trees of a given taxon, we also calculated the proportion of points in each grid cell with no trees. For taxon-specific analysis, we calculated the proportion of points with no trees of the taxon of interest.

Traditionally, basal area at an aggregated level has been calculated as the product of the mean density and mean tree basal area, but because of their negative correlation, this traditional approach overestimates the average values [37–39]. Our estimates of biomass and basal area in a grid cell avoid this problem by instead calculating the mean of the point-level multiplication of density and tree size [28]. Similarly, our estimate of density for a given taxon is calculated as the average of the point-level density estimates for that taxon. Furthermore, the biomass of each taxon is the mean of the point-level biomass estimates.

**Statistical smoothing.** The major challenge of modeling stem density, basal area, and biomass data is that these quantities are both non-negative and continuous, with a discrete spike at zero (i.e., "zero inflated"); few statistical distributions are available for this type of data. The description below is specifically for biomass for concreteness and clarity of presentation, but the modeling structure is the same for basal area and stem density.

There are many zero-inflated models in the statistical literature, most focusing on count or proportional data [40,41]. In early efforts we considered a Tweedie model [42,43]. However, computational difficulties affected model convergence, and the Tweedie model produced poor fits. Given this, we developed a two-stage model to address the challenge of zero inflation in non-negatively valued distributions. Our model was motivated by the biological insight that local conditions may prevent a taxon from occurring locally even though the taxon may be present at high density nearby. Thus, we combine a model for "potential biomass", which reflects the large-spatial-scale patterns in biomass with a model for "occupancy", which reflects the propensity for a given forest stand to contain the taxon. This model allows for zero inflation because a low probability of occupancy can easily produce observations that are zero at the grid cell aggregation.

Let $N(s)$ be the number of PLS sample points in grid cell $s$. Let $n_p(s)$ be the number of points in grid cell $s$ that have one or more trees of taxon $p$. Let $\bar{Y}_p(s)$ be the average biomass for taxon $p$ calculated only from the $n_p(s)$ points at which the taxon is present. In other words, $\bar{Y}_p(s) = \frac{1}{n_p(s)} \sum_{i=1}^{N(s)} Y_{ip}(s)$ where $i$ indexes the $N(s)$ sample points within cell $s$. Let $m_p(s)$ be the potential biomass process (we consider it both on the log scale and the original scale) and $\theta_p(s)$ be the occupancy process, both evaluated at grid cell $s$. The biomass in a grid cell can then be calculated as $b_p s = \theta_p(s) \, exp\,(m_p(s))$, (for the case when $m_p(s)$ is modelled on the log scale) namely weighting the average biomass in "occupied points" by the proportion of points that contain the taxon.

First consider the occupancy model. The likelihood is binomial, $n_p(s) \sim Bin(N(s), \theta_p(s))$. Note that the occupancy model represents the occupancy of points within a grid cell for taxon $p$ and that $\sum_p \theta_p(s) > 1$ because two taxa will often "occupy" the same point, since most PLS points have two trees. Next consider the (log) biomass process. We considered modeling potential biomass both on the original scale, $\bar{Y}_p(s) \sim N\left(m_p(s), \frac{\sigma_p^2}{n_p(s)}\right)$, and on the log scale, $log\,\bar{Y}_p(s) \sim N\left(m_p(s), \frac{\sigma_p^2}{n_p(s)}\right)$, where the scaling of the variance by $n_p(s)$ is the usual variance of an average. This likelihood accounts for heteroscedasticity related to the number of points at which the taxon is observed (not the number of PLS points in the cell). Finally, for total (non-taxon-specific) biomass, $n_p(s)$ is simply the number of points with any trees.

We found that working on the log scale produced more accurate point and uncertainty estimates based on cross-validation (S3 Appendix). This improved performance likely results from 1) downweighting the influence of outliers and 2) the log-scale model inherently having its variance scale with the mean (when both are considered on the original scale), which we observe empirically in the raw grid-level data. A disadvantage of using the log scale rather than the original scale (akin to using the geometric rather than arithmetic mean) is that by smoothing on the log scale, our estimates have a downward bias when transformed back to the original scale. This occurs because outlying values are discounted on the log scale, whereas on the original scale, outlying values have a strong influence through the squared error loss inherent in the likelihood. We also note that, based on the delta method [44], the correct approximate distribution when working on the log scale is $log\,\bar{Y}_p(s) \sim N\left(m_p(s), \frac{\sigma_p^2}{m_p(s)n_p(s)}\right)$. There is no clear means of accounting for the extra $m_p(s)$ in the denominator when fitting the potential biomass

on the log scale using generalized additive modeling (GAM) software (see below). We briefly considered using gamma regression (where the potential biomass is taken to be distributed according to a gamma distribution) in order to remain on the original scale, but we encountered computational difficulties in doing so with this large a dataset when using the GAM software discussed below. Hence, we employed the log-scale model while accepting some downward bias in the resulting estimates.

This two-stage model is able to account for the zero inflation produced by structural zeros (the taxon is not present because local conditions prevent it) through the use of the occupancy model. Through the potential model, it is also able to capture the smooth larger-scale variation in biomass. And by having both component models, we can account for the differential amounts of information caused by the large number of zeros and different numbers of sampling points in each grid cell.

Note that $m_p(s)$ is likely to be quite smooth spatially, at least for the PLS data, because when a point is occupied by a given taxon, the tree is likely to be of adult size, regardless of whether the tree is common in the grid cell. Thus, most of the spatial variation in biomass may be determined by variability in occupancy. The potential biomass is meant to correct for the fact that density and tree size may vary somewhat, but probably not drastically, across the domain.

We fit the two-component models using penalized splines to model the spatial variation, with the fitting done by the numerically-robust GAM methodology implemented in the R package mgcv [45]. We use the GAM implementation intended for large datasets encoded in the bam() function [46] in place of the usual gam() function.

The GAM methodology determines a data-driven compromise between simply using the noisy grid-level estimates (in which much of the spatial variation is statistical noise) and averaging over large spatial regions (which prevents seeing real finer-scale spatial variation). Internally during the fitting, the methodology uses a form of cross-validation (separate from the cross-validation results reported in S3 Appendix) to determine the optimal degree of spatial averaging that smooths over noise while retaining spatial signal. As can be seen in Results, this spatial averaging smooths out sharp spatial variations (likely from statistical noise) at the scale of 1–5 grid cells (several townships) while resolving larger-scale spatial features.

We accounted for the heterogeneity in the number of occupied points per grid cell by setting the 'weights' argument in the bam() function to equal to $n_p(s)$. We also considered scaling all weights by dividing by 70, where 70 is the approximate number of points in a cell that was fully surveyed. This treats a fully-covered cell as having one 'unit' of information and scales the contribution to the likelihood from cells with a different number of points relative to that. However, the results with and without the division by 70 were identical for the point estimates and very similar for the uncertainty estimates, so our final results omit this scaling.

We did not use environmental covariates as predictors in our statistical model for several reasons. First, we have fairly complete coverage (see Fig 1), such that the use of covariates is expected to provide limited additional information. Second, covariates such as climate for the settlement time period are not available, and we were reluctant to assume that present-day values are sufficiently similar to values in the past. Finally, without developing complicated statistical models that allow the effect of covariates to vary spatially (so-called varying-coefficient models), using regression coefficient estimates that are constant spatially can cause biases, such as inferring the presence of a taxon outside of its range boundary. For these reasons, we chose to rely only on spatial smoothing of the raw data. In the future, researchers could use our raw data products in combination with covariates.

Finally, to estimate total biomass, we fit the model above to raw total biomass values from the survey points, aggregating in the same fashion as described above for individual taxa, but including data from all trees. We note that the estimated total biomass at individual grid points

is systematically higher than summing over the taxon-specific estimated values; this results from smoothing on the log scale, which more strongly discounts the more extreme taxon-specific values in individual grid cells than would be the case if working on the original scale.

**Quasi-Bayesian uncertainty estimates.** As discussed in [45], one can derive a quasi-Bayesian approach and simulate draws from an approximate Bayesian posterior by drawing values of the spline coefficients based on the approximate Bayesian posterior covariance provided by gam() or bam() and, for each draw, calculating a draw of $\theta_p(s)$ and similarly a draw for $m_p(s)$ for the biomass process. We combined 250 draws from the occupancy and potential biomass processes (assuming independence between the processes) to produce biomass draws for each taxon and for total biomass. The procedure for stem density and basal area was analogous.

Note that one major drawback of our methodology of fitting the taxon-specific models separately is that individual taxon estimates are not constrained to add to the total biomass values estimated from using our model on raw total biomass values because the taxa are fit individually. Further, as in our related modeling of composition [22], we do not capture correlations between taxa, in part to reduce computational bottlenecks and in part to avoid inferring the value of one taxon based on the value of another. While there are real correlations, the correlation structure likely varies substantially over space (e.g., two taxa that positively covary can have different range boundaries such that the presence of one beyond the boundary of another does not indicate the presence of the second taxa). Since information is available for all taxa at any location with data, there is little need to borrow strength across taxa to improve our point estimates (unlike the need to borrow strength across space to fill in missing areas and smooth over noise caused by limited data in each grid cell). However, since the taxa are fit independently of each other, but are not truly independent, one cannot estimate uncertainty in any quantities that are functions of two or more taxon-specific estimates. Also note that one might scale the taxon-level point estimates to add to the total estimates, but there is no clear way to do this at the level of the posterior draws, because the draws are computed independently between the total and taxon-specific fits. In summary, we are able to calculate uncertainties in the total biomass and in the biomasses for individual taxa but unable to calculate uncertainty for any variable that combines two or more of the taxon-specific variables.

In the GAM fits, we noticed some anomalies in the quasi-posterior draws for the occupancy model that were likely caused by numerical issues. In particular, for some taxa, there were draws of the occupancy probability that were more than five times as large as the corresponding point estimate of the probability. Most of these occurred for very low occupancy probabilities in areas outside the apparent range boundary for the taxa. There were also cases where draws of total (non-taxon-specific) occupancy probability were near zero even though the probability point estimate was essentially one. As an ad hoc, but seemingly effective solution, we made the following adjustments to the draws:

1. set draws where the taxon-specific occupancy probability is greater than five times the point estimate to be equal to the point estimate, and

2. set all draws where the total point estimate was greater than 0.999 to be equal to 1.

**Choice of smoothing and scale of averaging.** We used cross-validation for total biomass, basal area, and stem density, as well as on a per-taxon basis, at the grid scale to:

1. choose between estimating potential biomass on the log-scale or original scale, and

2. determine the maximum number of spline basis functions, denoted as $k$.

With regard to the maximum number of basis functions, while the generalized additive modeling methods of [45] choose the amount of smoothing based on the data, using a large number of basis functions can result in slow computation. We hence chose to limit the number of basis functions and thereby impose an upper limit on the effective degrees of freedom of the spatial smoothing estimated from the data. The choice of that limit was informed by cross-validation. However, the imposed upper limit to the number of possible basis functions was large enough to have little effect on the amount of smoothing, although possibly imposing slightly more smoothing than without the limitation.

We used 10-fold cross-validation, randomly dividing the grid cells into 10 sets and holding out each set in turn. This allows us to assess the ability of the model to estimate biomass for cells with no data (and also gives us a good sense of performance for cells with very few points). Note that even with our incomplete sampling in Indiana and Illinois (Fig 1, 65% and 42% of the total area of each state, respectively), most unsampled grid cells are near to other grid cells with data. We considered values of $k$ in the set {100, 250, 500, 1000, 1500, 2000, 2500, 3000, 3500}.

The metrics used in cross-validation were absolute error loss for the point predictions relative to the grid-level raw data and statistical coverage of prediction intervals of the grid-level raw data.

We calculated absolute error weighted by the number of PLS points in the held-out cell and truncated both held-out values and predictions to maximum values of 600 Mg/ha to avoid having very large values overly influence the assessment. This also allowed us to work on the original (not log) scale in our evaluation, as we didn't want to accentuate small differences at low biomass values when choosing amongst modeling approaches, although we did end up choosing to fit the model on the log scale based on this empirical evaluation.

We calculated 90% prediction interval coverage using a modified version of the quasi-Bayesian uncertainty procedure described previously. The modification to the sampling procedure involves drawing a random binomial value based on each draw of the occupancy probability, multiplied in turn by a random normal draw (exponentiated when fitting the model on the log scale) centered on the draw of the potential surface with variance equal to the residual variance from the potential model. The addition of the binomial draw and the residual variance produces a prediction interval for the data rather than the unknown process and allows us to assess coverage relative to an observed quantity. We calculated 90% prediction intervals using the 5th and 95th percentiles of the 250 draws for each held-out cell. Coverage was determined as the proportion of cells for which the observation fell into the interval, considering only grid cells with at least 60 PLS points. We also calculated the median length of intervals (and median log-length) to assess the sharpness of the intervals, as high coverage can always be trivially obtained from overly-wide intervals.

Unfortunately, the coverage results cannot directly assess the uncertainty estimates provided in the gridded data product. This is because the true biomass is unknown and thus cannot be used to judge coverage. We can only judge coverage of prediction intervals for the data. Thus, under- or over-estimation of uncertainty for the true quantities may be masked by compensating over- and under-estimation of the residual error of the data around the truth.

Based on the cross-validation, we chose to use the models where the potential biomass (or stem density or basal area) is fit on the log scale, as this approach produced lower absolute error loss and gave a much better tradeoff between coverage and interval length (S3 Appendix).

**Data product.** We provide the following data products via the LTER Network Data Portal.

- Raw gridded stem density, aboveground biomass and basal area at https://portal.lternet.edu/nis/mapbrowse?scope=msb-paleon&identifier=26&revision=1 (DOI: 10.6073/pasta/801601af769fa5acade1ef07f6892bdd).

- Gridded statistically-smoothed stem density at https://portal.lternet.edu/nis/mapbrowse?scope=msb-paleon&identifier=24&revision=0 (DOI: 10.6073/pasta/1b2632d48fc79b370740a7c20a70b4b0).

- Gridded statistically-smoothed aboveground biomass at https://portal.lternet.edu/nis/mapbrowse?scope=msb-paleon&identifier=23&revision=0 (DOI: 10.6073/pasta/b246e05afb25dbe06b3006c5d18a4a2b).

- Gridded statistically-smoothed basal area at https://portal.lternet.edu/nis/mapbrowse?scope=msb-paleon&identifier=25&revision=0 (DOI: 10.6073/pasta/c3ae2363e4ae2e0f42a7c02b6f12b50a).

  We also provide point-level raw data:

- Indiana - https://portal.lternet.edu/nis/mapbrowse?scope=msb-paleon&identifier=27&revision=0 (DOI:10.6073/pasta/c3e2404f5b34204b5871a743ebce3c51)

- Illinois - https://portal.lternet.edu/nis/mapbrowse?scope=msb-paleon&identifier=28&revision=0 (DOI: 10.6073/pasta/b8fdabd7cfba3a2b3d55fe4c1dc5383f)

- Michigan -

  ○. Southeastern - https://portal.lternet.edu/nis/mapbrowse?scope=msb-paleon&identifier=29&revision=0 (DOI: 10.6073/pasta/409ec6dfb218b6a3e98022916d2b4438)

  ○. Southern - https://portal.lternet.edu/nis/mapbrowse?scope=msb-paleon&identifier=30&revision=0 (DOI: 10.6073/pasta/8d033c1cfadca42bf060f9f38940c81e)

  ○. Northern - https://portal.lternet.edu/nis/mapbrowse?scope=msb-paleon&identifier=31&revision=0 (DOI: 10.6073/pasta/3760eec82562e0a8b7cd493c0a3e3ef4)

- Wisconsin - https://portal.lternet.edu/nis/mapbrowse?scope=msb-paleon&identifier=32&revision=0 (DOI: 10.6073/pasta/c3e680e51026e74a103663ffa16cb95d)

- Minnesota - https://portal.lternet.edu/nis/mapbrowse?scope=msb-paleon&identifier=33&revision=0 (DOI: 10.6073/pasta/f55f6b7f4060a9b4f07374e7db8443cd)

  The project GitHub repository (https://github.com/PalEON-Project/PLS_products) provides code for processing the point-level data and producing the data products above in the subdirectory named 'R'. In the subdirectory 'data/conversions', we provide:

- our translation tables for translating surveyor taxon abbreviations to modern common names, including aggregation for the raw gridded values and statistical modeling done in this work,

- correction factors for the subregions of the domain for estimating point-level tree density, and,

- our assignments of allometric relationships for the PalEON taxa, based on [29].

## Results

### Presettlement vegetation structure and biomass: Spatial variation and regional averages

Across our domain there are large variations in estimated total stem density, basal area, and aboveground biomass, and the smoothed estimates reveal spatial patterns that can be hard to

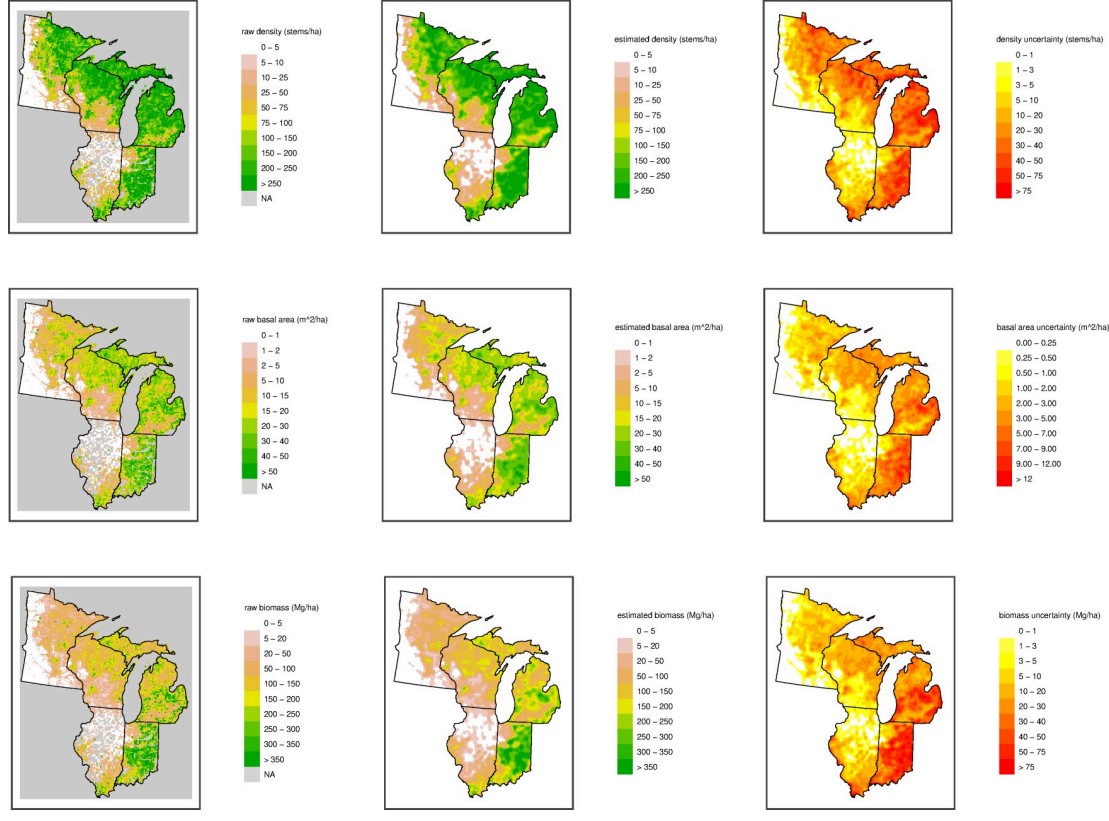

**Fig 2. Spatial patterns of vegetation structure.** Raw data, predictions and uncertainty for total stem density (stems/ha; top row), total basal area (m²/ha; middle row), and total biomass (Mg/ha; bottom row). Point estimates are from raw data in each cell based on the average point-level biomass (left column), predictions are estimates from the statistical smoothing model (middle column), and uncertainty estimates are from the standard deviation of the quasi-Bayesian posterior draws (right column). Note that in the raw data plots, grey indicates data were not available for a grid cell (this occurs rarely, except in Illinois and Indiana).

see amidst the noisiness in the raw grid-level data (averaging over the point-level estimates) (Fig 2). Similarly large variations are also seen at the taxon level (Figs 3 and 4).

Because this paper is intended mainly to describe the methods and resulting data products, the results presented here focus on illustrating the potential for using these data to understand the patterns of variation in structure between regions and over time. Hence in the results that follow, we mostly report simple point estimates without uncertainty. However, the estimates with uncertainty are fully suitable for further and more formal statistical analyses.

The 8 km grid used here averages across a presettlement mosaic of forest types and integrates areas of disturbances, different ages, and peatlands. Thus, the structural estimates of density, basal area, and biomass reported in Fig 2 represent broad-scale landscape characteristics, not those found in forest stands or uniform forest types. Furthermore, there are three interdependent but distinct estimates of the structural variables: the average of empirical point values within each grid cell (64 km²); modeled and smoothed values that smooth over small-scale spatial variation and therefore represent variation at the scale of several townships (roughly 400 km²); and the sum across the individual taxa smoothed independently at the same multi-township scale (the individual taxon estimates are seen in Figs 3 and 4). The three estimates of biomass indicate different facets: the raw empirical estimate is an integrated mean across the landscape, the modeled values are the expectation at a moderate spatial scale across the whole domain, and the taxa total is the rescaling of the relative composition in the context

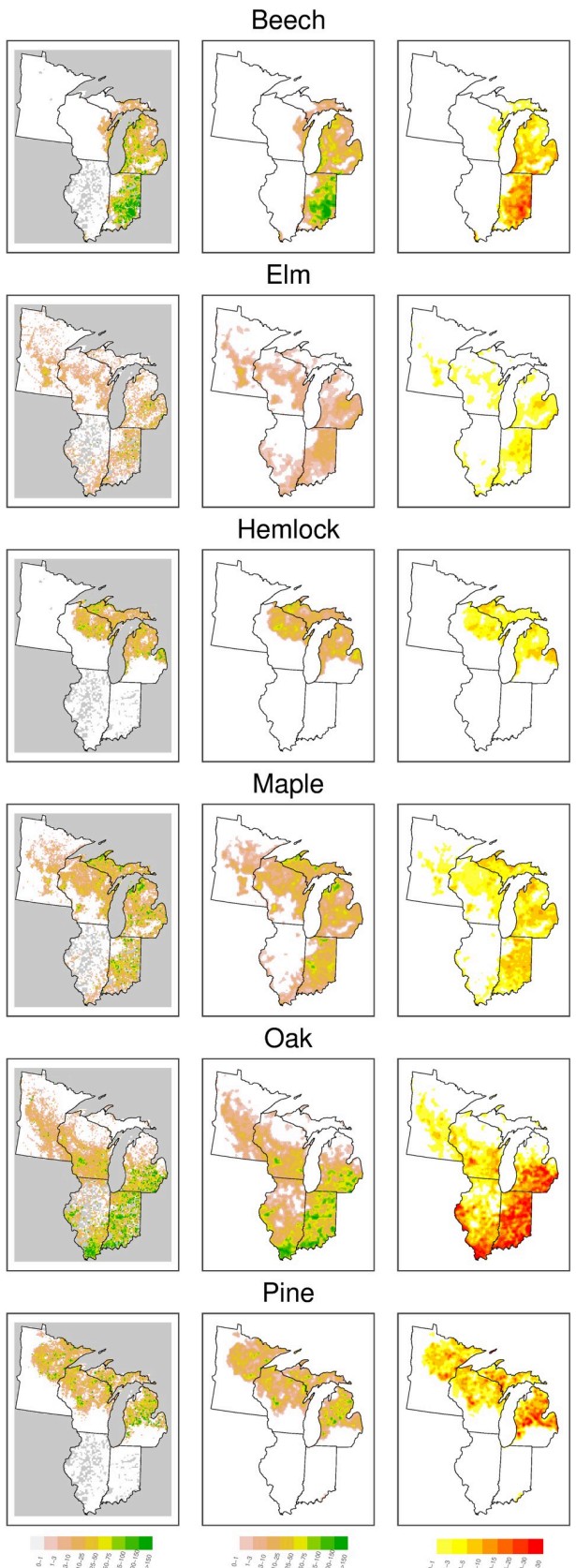

**Fig 3. Spatial patterns of biomass by for select data.** Raw data, predictions, and uncertainty of biomass (Mg/ha) for select taxa. Column ordering and figure formatting follow Fig 2.

of a patchy spatial distribution of taxa. Estimates of average forest structure (density, basal area, and biomass) across the study domain when based on raw values (160 trees/ha, 15 m$^2$/ha, 105 Mg/ha) are larger than the corresponding averages from the modeled values (137 trees/ha, 13 m$^2$/ha, 91 Mg/ha) (Table 1). These whole-domain averages, moreover, are unrepresentative of any particular forest or zone as they are a mixture of treeless grasslands, woodlands, and forest. In addition, raw values are not yet available for all of Illinois and Indiana, so these areas are underrepresented in the raw whole-domain average values. The smoothing also tends to downscale the maximum density values (because of the use of the log scale) and thus the models underfit the total density by about 14%. Basal area and biomass follow density and are

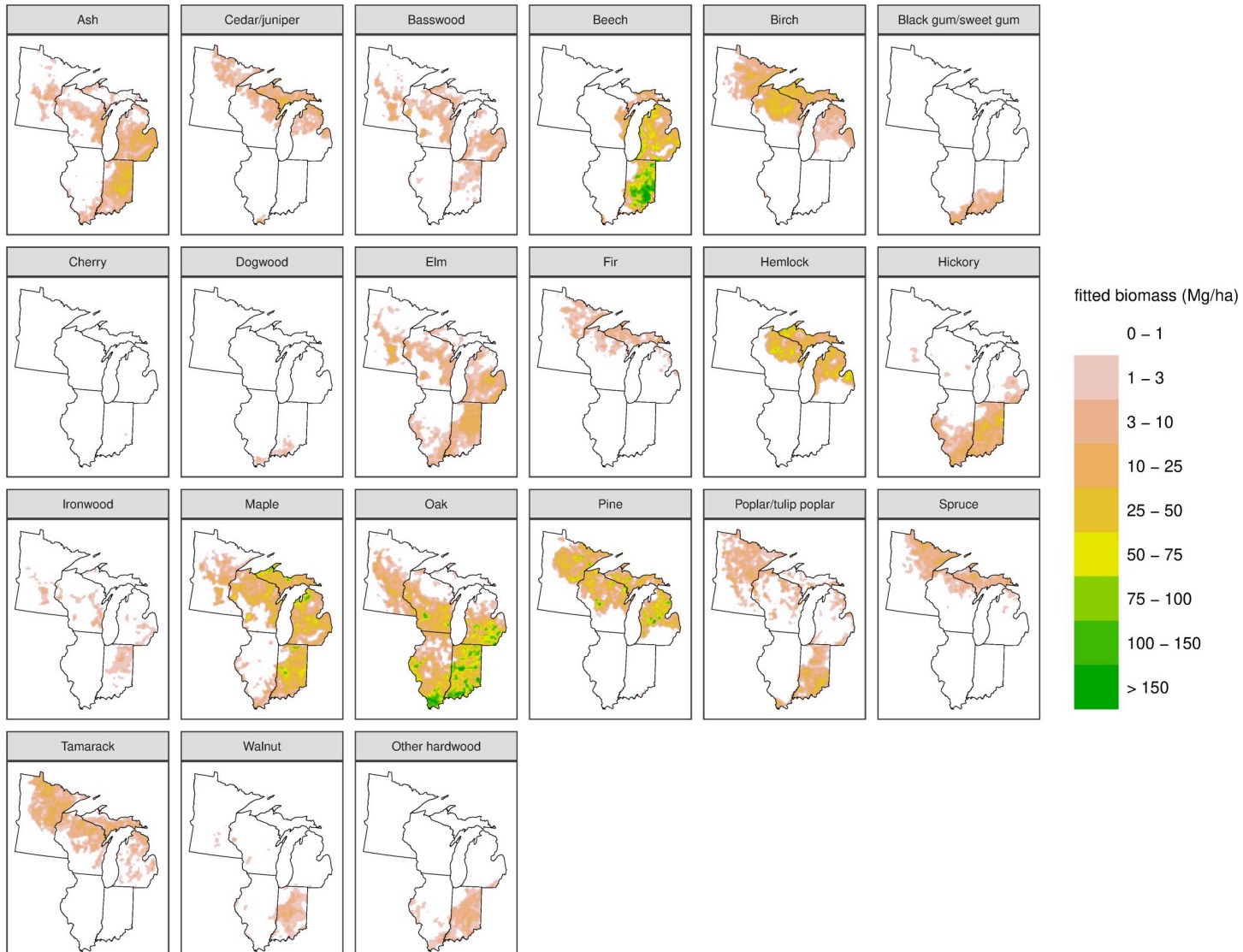

**Fig 4. Predictions of biomass (Mg/ha) from the statistical smoothing model for all taxa.**

**Table 1. Average presettlement raw and modeled structure and composition in geographic divisions (Fig 5) of the Midwest.**

| Zone | Empirical (Raw) [1] | | | | Modeled smoothed [2] | | | | | |
| | Cells [3] | Density | Basal Area | AGB [4] | Cells [5] | Density | Basal Area | AGB [4] | 95% -tile [6] | |
| | (n) | (stems >20cm/ha) | (m²/ha) | (Mg/ha) | (n) | (stems >20cm/ha) | (m²/ha) | (Mg/ha) | (Mg/ha) | Empirical Biomass Composition |
|---|---|---|---|---|---|---|---|---|---|---|
| W Minnesota | 2081 | 35 | 3.0 | 19 | 2097 | 29 | 2.5 | 15 | 67 | 34% oak, 15% elm, 12% maple, 9% aspen |
| NE Minn. | 1490 | 201 | 14.3 | 71 | 1506 | 197 | 13.2 | 64 | 125 | 34% pine, 17% birch, 15% tamarack, 9% spruce |
| W UP Mich. | 522 | 293 | 27.5 | 159 | 467 | 275 | 26.4 | 151 | 214 | 24% maple, 19% hemlock, 19% birch, 16% pine |
| N Wisconsin | 1088 | 256 | 21.5 | 129 | 1020 | 243 | 20.7 | 123 | 172 | 18% birch, 17% hemlock, 16% pine, 16% maple |
| N Michigan | 921 | 286 | 22.6 | 132 | 809 | 268 | 21.7 | 126 | 200 | 22% pine, 18% maple, 16% beech, 16% hemlock |
| S Wisconsin | 1068 | 75 | 6.6 | 49 | 1062 | 68 | 6.0 | 43 | 126 | 51% oak, 13% maple, 9% pine |
| W Illinois | 1346 | 34 | 4.5 | 40 | 2330 | 27 | 3.5 | 30 | 105 | 68% oak, 9% hickory |
| S Michigan | 1089 | 228 | 25.0 | 194 | 1063 | 205 | 21.8 | 169 | 284 | 27% oak, 17% beech, 13% maple, 11% pine |
| E Indiana | 860 | 262 | 33.1 | 315 | 949 | 245 | 31.3 | 285 | 426 | 30% beech, 25% oak, 12% maple, 9% tuliptree |
| S Illinois-Ind. | 410 | 176 | 19.7 | 166 | 468 | 168 | 18.8 | 159 | 233 | 55% oak, 9% hickory, 8% beech |
| all | 10875 | 160 | 15.0 | 105 | 11771 | 137 | 12.9 | 91 | 250 | 22% oak, 13% beech, 13% maple, 11% pine |

[1] Mean value of cell-specific point estimates averaged over 8x8 km cells within zone.

[2] Mean value of statistical model fit of values over 8x8 km cells within zone.

[3] Only incudes 8x8 km cells with PLS data, thus incomplete grid.

[4] Aboveground biomass.

[5] 8x8 km cells within complete grid, differing from empirical because cells with no PLS data are modeled.

[6] 95th percentile of aboveground biomass across cells in zone.

underestimated by approximately the same amount (Table 1). Moreover, the modeling of individual taxa compounds the underestimate, due to the patchy distributions of less common taxa, so that the sum of the modeled taxon-specific estimates averages 80% of the modeled total values (based on fitting a simple linear regression). While the empirical totals are an unbiased (but noisy) estimate of the actual pool of biomass, the modeled values better represent the variation and uncertainty across the domain.

These maps (Fig 2) illustrate broad regional differences in forest structure. Structural variations closely parallel compositional gradients, with prairie or savanna to the west and closed-crown forests to the north and east. The structure (based on the smoothed values) varies across the region from savanna (27 Mg/ha) to northern hardwood (104 Mg/ha) and mesic southern forests (211 Mg/ha). To summarize these regional variations, we split the domain into ten contiguous regions of similar density and composition (Fig 5A, Table 1) and, as an alternative, 12 EPA Level 3 Ecoregions (Fig 5B, S1 Table). Two regions (western Minnesota and western Illinois) are widespread prairie with oaks the dominant tree genus (less than 50 trees/ha, 5 m²/ha, 40 Mg/ha) and one region (southern Wisconsin) is predominantly savanna (up to 100 trees/ha, 10 m²/ha, 60 Mg/ha). The meso-scale patterns of forest outliers in the prairie (Big Woods

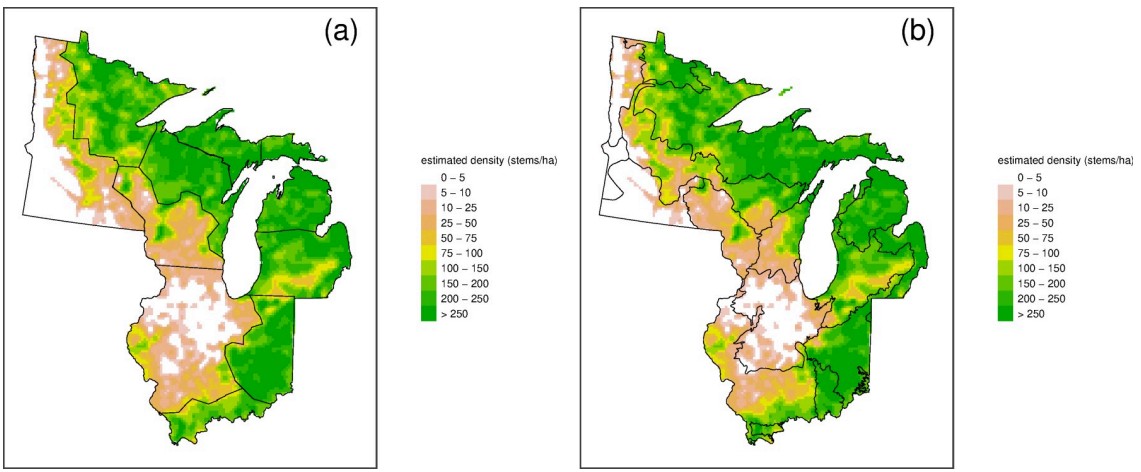

**Fig 5.** Modeled forest density and delineation of study domain into (a) 10 zones of similar structure and (b) EPA ecoregions.

of Wisconsin and Minnesota, and west of the Illinois River in west-central Illinois) are clearly distinguished as islands of higher density, basal area, or biomass.

Although the presettlement forests in different regions have similar densities, between 200 and 300 trees/ha (>20 cm dbh), they distinctly differ in their dominant species and tree sizes (Table 1). The pine-birch forests of the northwest (northeast Minnesota) had small stature (14 m$^2$/ha, 71 Mg/ha) consistent with persistent disturbance, especially fire [7]. The less-disturbed mixed northern hardwood-conifer forests in the north (northern Wisconsin, western Upper Peninsula and northern Michigan) had different mixtures of dominant species (birch, pine, maple, hemlock), but all had moderate stature (21–28 m$^2$/ha, 125–160 Mg/ha) (Fig 2, Table 1). Low-density savanna inclusions occur in forested areas as oak barrens, pine plains, or conifer swamps of northern Wisconsin and Michigan. Southeastern areas (southern Michigan, eastern Indiana) had well-developed forest (25–33 m$^2$/ha, 194–315 Mg/ha) of mixed hardwoods (beech, maple, oak) interspersed with hardwood swamps. A distinct salient of oak savanna extended across southern Michigan, bisecting the mesic hardwoods, lowering the regional average structure and adding a substantial oak component. The presettlement landscape reached a maximum forest structure in the mixed mesophytic forest of eastern Indiana, with pockets of forest averaging up to 55 m$^2$/ha and 600 Mg/ha at the township-scale (100 km$^2$) (Fig 2). In the south (southern Illinois and southern Indiana) the forests transition to more open and smaller-stature oak-hickory dominated landscapes (176 trees/ha, 20 m$^2$/ha, 166 Mg/ha).

## Changes in biomass and forest structure from presettlement to present

These PLS-based statistical estimates of presettlement forest biomass can be augmented by several independent lines of evidence: modern remnants, computer models, characteristics of current forests in the same location, and historical case studies in the literature. These analogs are highly variable but generally predict biomass values greater than 150 Mg/ha with a ceiling of 350 Mg/ha for old-growth stands, models, or current forest maximum stature (Table 2). These estimates are much higher than our estimates of a landscape biomass of about 100 Mg/ha in the Midwest. Some of these discrepancies can be attributed to widely different assumptions, methodologies, and the spatial grain and extent of analysis. Much of this dichotomy is due to the lack of clear analogs: models are simplistic and theoretical; old growth remnants have unique histories and are not representative of the broad landscape; modern forests are the

**Table 2. Analogs, reference, and context for biomass in the Midwest.** Values and ranges (in parentheses) are across zone-, forest-, stand-, or model-specific values.

| Presettlement | Source | Sample | AGB[1] (Mg/ha) | BA[2] (m2/ha) |
|---|---|---|---|---|
| Savanna | this study | 3 zones[3] | 35 (22–49) | 4.8 (3.9–6.6) |
| Northern Hardwood | this study | 4 zones[3] | 117 (77–151) | 20.6 (15.4–27.5) |
| Southern Mesic Hardwd. | this study | 3 zones[3] | 233 (166–315) | 27.0 (19.7–33.1) |
| all weighted | this study | 10 zones | 103 (22–315) | 14.6 (3.9–33.1) |
| Presettlement Literature | | | AGB (Mg/ha) | BA (m²/ha) |
| Northern Wisconsin | [2] | | 115 | |
| Southern Wisconsin | [2] | | 59 | |
| Central Minnesota | [47] | | 72 | |
| Northeastern Minnesota | [47] | | 38 | |
| Minn., Wisc., Mich. | [7] | | 109 | 23.1 |
| Modern Old Stands | | Surveys | AGB (Mg/ha) | BA (m²/ha) |
| Upper Peninsula, Mich. | [48–51] | 3 forests | 319 (260–411) | 41.9 (38.9–44.0) |
| Mich., Wisc., Minn. | [28] | 8 hardwood stands | 286 (207–381) | 37.6 (31.1–46.2) |
| Mich., Wisc., Minn. | [52] | 129 hardwood stands | | 33.6 (14.4–69.1) |
| Wisc., Mich. | [52] | 53 hemlock-pine stands | | 47.2 (20.0–76.6) |
| Indiana, Illinois | [53] | 14 oak stands | 234 (117–342) | 33.5 (28.1–39.5) |
| Indiana, Illinois | [52] | 33 oak stands | | 28.6 (11.8–47.2) |
| Process Models Spin-up | | | AGB (Mg/ha) | |
| Michigan Upper Peninsula | PALEON, [51] | 6 models [4] | 208 (106–322) | |
| Eastern Indiana | PALEON | 4 models[5] | 131 (109–158) | |
| Modern (FIA)[6] | | | AGB (Mg/ha) | BA (m²/ha) |
| Savanna | FIA | 3 zones[3] | 138 | 17.6 |
| Northern Hardwood | FIA | 4 zones[3] | 84 | 24.1 |
| Southern Mesic Hardwd. | FIA | 3 zones[3] | 131 | 24.9 |
| Old-growth region UP MI | FIA | North. hardwood | 155 (125–189) | 34.8 (27.7–39.5) |
| all weighted | FIA | 10 zones | 110 | 22.2 (19.5–24.9) |

[1] Aboveground biomass based on averaging the empirical values.

[2] Basal area based on averaging the empirical values.

[3] Geographic regions from Fig 5. Savanna includes western Minnesota, southern Wisconsin, and western Illinois. Northern hardwood includes northeastern Minnesota, western Upper Peninsula. of Michigan, northern Wisconsin, and northern Michigan. Southern mesic hardwoods includes southern Michigan, eastern Indiana, and southern Illinois/Indiana.

[4] Models: ED2; LPJ-GUESS; LPJ-WSL; LINKAGES.

[5] Models: ED2; LPJ-GUESS; LPJ-WSL; LINKAGES; TRIFFID; CANOPY.

[6] FIA data provided to PalEON under US Forest Service MOU.

result of long human management and they developed in different environments including climate and fire regimes; and historical case studies are spatially limited and not representative of a heterogeneous landscape. Surprisingly the gross average presettlement biomass (91 Mg/ha) is very similar to the modern forests across the Midwest domain (110 Mg/ha) (Table 3). This is misleading, however, since the modern US Forest Service Forest Inventory and Analysis (FIA) values are derived from only forested plots while the presettlement values include substantial areas of prairie or low-density woodlands. If primarily forested regions are considered, the weighted average presettlement biomass is close to 156 Mg/ha. This is about 50% higher than modern landscape values, but less than half the biomass that would occur if current old-growth forests covered the area (234–286 Mg/ha, Table 2).

**Table 3. Comparison of presettlement (based on smoothed values) and modern density and aboveground biomass over 10 regions of the Midwest.**

| Zone | | Presettlement (PLS) | | Modern (FIA) | |
|---|---|---|---|---|---|
| | n | density (stems/ha) | AGB (Mg/ha) | density (stems/ha) | AGB (Mg/ha) |
| Western Minnesota | 2097 | 29 | 15 | 158 | 110 |
| Northeast Minnesota | 1506 | 197 | 64 | 159 | 56 |
| Western UP MI | 467 | 275 | 151 | 220 | 94 |
| Northern Wisconsin | 1020 | 243 | 123 | 195 | 89 |
| Northern Michigan | 809 | 268 | 126 | 212 | 85 |
| Southern Wisconsin | 1062 | 68 | 43 | 172 | 103 |
| Western Illinois | 2330 | 27 | 30 | 164 | 145 |
| Southern Michigan | 1063 | 205 | 169 | 179 | 113 |
| Eastern Indiana | 949 | 245 | 285 | 181 | 147 |
| Southern Illinois-Indiana | 468 | 168 | 159 | 190 | 146 |
| Savanna[1] | 5489 | 36 | 27 | 163 | 123 |
| Northern Hardwood[1] | 3802 | 235 | 104 | 187 | 76 |
| Southern Mesic Hardwood[1] | 2480 | 213 | 211 | 182 | 132 |
| All | 11771 | 138 | 91 | 175 | 110 |

[1]See footnote 3 of Table 2 for definitions of these regions.

The moderately low biomass values imply that the presettlement landscape was not covered with forests of the stature of classic old modern stands, but a mixture of younger disturbed areas, habitats such as peatlands or sand plains with lower biomass, and well-developed forests of various compositions. Given highly variable presettlement raw data from a few trees at each point separated by at least 0.8 km, it is hard to distinguish the occasional high biomass stands. The northern forest reached peak presettlement development in the western Upper Peninsula of Michigan (average 28 m$^2$/ha, 95% of cells less than 214 Mg/ha) (Table 1), which is substantially less than the average 42 m$^2$/ha and 319 Mg/ha in three modern old growth forests in the region (Table 2). It is likely that the maximum point values were similar to modern old pine or hemlock stands.

The transition (tension zone) from conifer-northern hardwoods to smaller-stature oak forests seen in Wisconsin [54] is repeated in Michigan. A second center of hardwood forest, however, occurred in eastern Indiana and Ohio, which formed a gradual transition into oak forest to the south. The mesic hardwoods of the Lower Peninsula of Michigan and those in Indiana/Ohio are separated by the oak salient on sandy outwash soils in southern Michigan [21]. The hardwood forests in Wisconsin and Michigan (136 Mg/ha, 22 m$^2$/ha) were substantially smaller in stature than those in eastern Indiana (315 Mg/ha, 33 m$^2$/ha). There was a distinct north-south gradient with increasing biomass to the south. The presettlement beech-maple forests of Indiana were more southern in character with mixed mesophytic species such as tulip tree and had a regional biomass equivalent to modern old-growth stands (234 Mg/ha, 34 m$^2$/ha; Table 2). This region apparently supported forests with nearly old-growth structure across the presettlement landscape. The most massive Midwestern presettlement forests approached the size of the mixed-mesophytic old-growth stands (353 Mg/ha) of the coves in the southern Appalachian Mountains [55].

Although the biomasses of the presettlement and modern forests averaged over the Midwest are very similar, the changes in biomass after settlement have been different in different regions. Areas of prairie or savanna have gained aboveground biomass, at least in those areas now forested today, while forested areas have maintained their structure as original natural

disturbances have been replaced by human disturbance (Table 3). The two distinct presettlement hardwood regions of the Midwest (northern Wisconsin and Michigan versus southern areas of Indiana and Ohio) have different trajectories into the present. The massive mesic mixed hardwoods in southern areas have been converted to oak woodlots with decreased biomass. Despite a long history of logging producing many areas of younger stands, including shade-intolerant species, the northern areas have not substantially changed composition or biomass over time.

Estimates of structural properties of forests from sparse PLS point data are highly variable and using the Morisita estimator for density requires at least 600 points for an accurate determination [28]. In this study we agglomerate estimates into a grid (~70 points per cell) and then smooth over several townships to gain an adequate sample size for density determination. Sparse sampling of a mosaic of forest types leads to even more uncertainty, so the landscape averages are more informative than a prediction at any particular point. Additionally, the standard deviation of the grid values is strongly correlated with the mean values (Fig 2). Thus, the coefficient of variation of the density is fairly predictable (CV 90–140% for savanna and 25–40% for forested areas, S1 Table). Since basal area and biomass are the product of density and tree size, they display closely congruent spatial patterns of structure. Although biomass is a basic ecologically-significant property of forests, it requires more assumptions, is more complicated to calculate, and contains more uncertainty than density or basal area. Therefore, trends in all three parameters are important aspects of forest structure.

## Discussion

We have presented high-resolution estimates, with uncertainty, of stem density, basal area, and biomass at the time of Euro-American settlement for a large area of the midwestern United States. These estimates can be used to answer various questions about the patterns and processes governing forest composition and structure, as validation datasets for terrestrial ecosystem models, and as a baseline for understanding changes in ecosystems, including carbon storage, under anthropogenic change.

The presettlement landscape of the Midwest supported multiple dominant species, vegetation types, forest types, and ecological formations. The prairies, oak savanna, and forest each had distinctive structures and spatial patterns across the domain. Analysis of the early land survey records clearly quantify the structure of these divisions. The landscape averages of structure variables for the presettlement forests are greater than the modern highly-disturbed and harvested forested landscape, but substantially less than undisturbed modern remnants. The forests of northern Wisconsin and Michigan were of moderate average stature (275 trees/ha, 23 m$^2$/ha, 136 Mg/ha), while those in southern Michigan and eastern Indiana were more robust (243 trees/ha, 29 m$^2$/ha, 263 Mg/ha). The presettlement forests were neither unbroken and massively-statured nor constantly opened by natural disturbances. Overall, the forests were structurally between modern second growth and old-growth, but compositionally and visually similar to large segments of the modern landscape. The open savannas and prairies of the presettlement landscape, by contrast, have been almost completely replaced by agricultural land and medium density forests.

While our estimates have a variety of strengths, including relatively high resolution, relatively uniform data density, coverage of a large area, careful data cleaning, and the use of statistical methods tailored to the data, there are of course limitations. The 8-km grid resolution prevents study of variation at finer scales such as the stand level and from smaller scale effects such as local topography, including the effects of small fire breaks. For example, our total biomass and stem density estimates show a portion of the Minnesota River valley in southwestern

Minnesota (see Fig 2), but they cannot resolve riparian forest (relative to grassland or upland forests) in smaller valleys. Our estimates smooth over the local variation, which can include sharp ecotone boundaries. In future work in this and other domains, we plan to make use of the point level data without initial gridding to try to estimate finer-scale variation, although one will always be limited by the natural resolution of the PLS survey points.

Our statistical model cannot represent range boundaries as it models variation in abundance as a continuously-valued spatial field with strictly positive (but often negligibly above zero) predicted stem density, basal area, and biomass, compounded by the smoothing mentioned above. Of course, range boundaries are generally fuzzy.

Our statistical model fits each taxon separately, for computational convenience and to limit the complexity of the spatial statistical models. Thus, the uncertainty estimates do not capture any correlated uncertainty across taxa and analyses that aggregate estimates across more than one taxon (such as comparing two taxa or summing across multiple taxa) will not be able to correctly characterize uncertainty. For sums, one could, as we have done for stem density, basal area, and total biomass, sum the raw values and then apply the spatial statistical model. Finally, the sum across taxa of the taxon-specific estimates for a grid cell do not add to the estimate of total stem density, basal area, or biomass for that grid cell.

In this work, as in [22], we chose not to use environmental covariates, such as soils, firebreaks, and topography [4,56], when estimating stem density, basal area, and biomass. Instead we limited our model to capture variation solely based on smoothing the data using spline-based techniques that rely on spatial distances. This avoids dependence on the environmental drivers of presettlement forest composition that might cause circular reasoning in subsequent analyses that use our data products. In addition, use of covariates could also lead to prediction that a taxon is present well beyond its range boundary in places where data are sparse.

The estimates and raw data are available as public data products, and our methods are fully documented with code available in our GitHub repository.

## Supporting information

**S1 Table. Modeled density and aboveground biomass in the presettlement and modern periods over 10 regions and 12 ecoregions of the Midwest.**
(PDF)

**S1 Appendix. Data collection and cleaning.**
(PDF)

**S2 Appendix. The Morisita plotless density estimator and correction factors.**
(PDF)

**S3 Appendix. Model selection using cross-validation.**
(PDF)

## Acknowledgments

The authors are deeply indebted to all of the researchers over the years who have preserved, collected, and digitized survey records, in particular Robert McIntosh (deceased; formerly at University of Notre Dame), Ed Schools (Michigan State University Extension—Michigan Natural Features Inventory), and Ted Sickley (formerly at University of Wisconsin). We thank University of Wisconsin (Madison) undergraduates Madeline Ruid, Benjamin Seliger, Morgan Ripp and Daniel Handel for processing of the southern Michigan data and the Map Library in the Department of Geography at the University of Wisconsin for digitization of the Mylar

maps. Indiana and Illinois data were made possible through the hard work of over 30 University of Notre Dame undergraduates in the McLachlan lab. We thank Simon Goring for early work on data preparation and analysis. We thank Jun Zhu, Xiaoping Feng, and Wesley Brooks for early work on the spatial statistical methods presented here.

## Author Contributions

**Conceptualization:** Christopher J. Paciorek, Charles V. Cogbill, John W. Williams, Jason S. McLachlan.

**Data curation:** Charles V. Cogbill, Jody A. Peters, John W. Williams, David J. Mladenoff.

**Formal analysis:** Christopher J. Paciorek, Charles V. Cogbill.

**Funding acquisition:** Christopher J. Paciorek, John W. Williams, Jason S. McLachlan.

**Investigation:** Christopher J. Paciorek, Charles V. Cogbill, Jody A. Peters, John W. Williams, Andria Dawson, Jason S. McLachlan.

**Methodology:** Christopher J. Paciorek, Charles V. Cogbill, Jody A. Peters, David J. Mladenoff.

**Project administration:** Christopher J. Paciorek, Jody A. Peters, John W. Williams, Jason S. McLachlan.

**Resources:** Jody A. Peters, John W. Williams, David J. Mladenoff, Jason S. McLachlan.

**Software:** Christopher J. Paciorek, Jody A. Peters, Andria Dawson.

**Supervision:** Christopher J. Paciorek, Jody A. Peters, John W. Williams, Jason S. McLachlan.

**Validation:** Charles V. Cogbill, Jody A. Peters, David J. Mladenoff, Andria Dawson.

**Visualization:** Christopher J. Paciorek, Charles V. Cogbill.

**Writing – original draft:** Christopher J. Paciorek, Charles V. Cogbill.

**Writing – review & editing:** Christopher J. Paciorek, Charles V. Cogbill, Jody A. Peters, John W. Williams, David J. Mladenoff, Andria Dawson, Jason S. McLachlan.

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
