## [Decision Letter · Decision Letter 0]

10 Dec 2020

PONE-D-20-31336

The forests of the midwestern United States at Euro-American settlement: spatial and physical structure based on contemporaneous survey data

PLOS ONE

Dear Dr. Paciorek,

Thank you for submitting your manuscript to PLOS ONE. After careful consideration, we feel that it has merit but does not fully meet PLOS ONE’s publication criteria as it currently stands. Therefore, we invite you to submit a revised version of the manuscript that addresses the points raised during the review process.

Please further clarify the methods section of the paper and make other minor revisions concerned by the referees

We look forward to receiving your revised manuscript.

Kind regards,

RunGuo Zang

Academic Editor

PLOS ONE

Additional Editor Comments:

The manuscript is generally well written and deliver some information which would be interested by a number of readers.Please further clarify the methods section of the paper and make other minor revisions concerned by the referees.

Journal Requirements:

2. We note that Figures 1-5 in your submission contain map images which may be copyrighted. All PLOS content is published under the Creative Commons Attribution License (CC BY 4.0), which means that the manuscript, images, and Supporting Information files will be freely available online, and any third party is permitted to access, download, copy, distribute, and use these materials in any way, even commercially, with proper attribution. For these reasons, we cannot publish previously copyrighted maps or satellite images created using proprietary data, such as Google software (Google Maps, Street View, and Earth). For more information, see our copyright guidelines: http://journals.plos.org/plosone/s/licenses-and-copyright.

2.1.    You may seek permission from the original copyright holder of Figures 1-5 to publish the content specifically under the CC BY 4.0 license. 

2.2.    If you are unable to obtain permission from the original copyright holder to publish these figures under the CC BY 4.0 license or if the copyright holder’s requirements are incompatible with the CC BY 4.0 license, please either i) remove the figure or ii) supply a replacement figure that complies with the CC BY 4.0 license. Please check copyright information on all replacement figures and update the figure caption with source information. If applicable, please specify in the figure caption text when a figure is similar but not identical to the original image and is therefore for illustrative purposes only.

Reviewers' comments:

Reviewer's Responses to Questions

**Comments to the Author**

1. Is the manuscript technically sound, and do the data support the conclusions?

Reviewer #1: Yes

Reviewer #2: Yes

2. Has the statistical analysis been performed appropriately and rigorously? 

Reviewer #1: I Don't Know

Reviewer #2: I Don't Know

3. Have the authors made all data underlying the findings in their manuscript fully available?

Reviewer #1: Yes

Reviewer #2: Yes

4. Is the manuscript presented in an intelligible fashion and written in standard English?

Reviewer #1: Yes

Reviewer #2: Yes

5. Review Comments to the Author

Reviewer #1: The authors applied a statistical modeling approach to smooth the settlement-era Public Land Survey (PLS) data and presented gridded 8 km-resolution data products of the estimated stem density, basal area, and biomass of tree taxa at Euro-American settlement of the midwestern United States during the middle to late 19th century. This work is interesting, and the datasets generated may useful for studying several other related questions as listed in the manuscript.

The authors have published several papers on the related topic and the manuscript is generally well written. In my opinion, the “Methods” part is too long and has many detailed information and description, it may be better to present the Methods in a concise way and move the detailed description into a supplementary material. In addition, there are a few points the authors need to clarify in their manuscript.

1. Line 51, give a concise description of the “Historical datasets”, what kinds of data are included in the “Historical datasets”.

2. Line 57, from line 53 to 57, the authors listed several aspects in which the datasets are potentially useful and could be applied to study those questions. I suggest the authors illustrate the applications of their dataset to those questions using proper examples.

3. Line 87, briefly describe the results obtained by [7], and explain the relationship between this work and [7], and highlight the novelty and significance of the current work.

4. Line 101, from line 92 to 101, the authors summarized the main idea of their manuscript. I suggest the authors clearly present the significance of their work at the end of the “Introduction”.

Reviewer #2: I found this paper fascinating. The authors' story telling ability is appreciated. The paper mede me realize the potential of spatial and statistical methods in teasing out information from the historic databases. While the paper involves complex statistical analysis and makes some big conclusions, but the authors have acknowledged and discussed well the limitations of the methods and the caveats of the results and conclusions. Not perfect, but given the nature of the data and the question of interest, the methods seem reasonable. Some of the statistical techniques used are beyond my knowledge of the subject, but I have no reason to believe that there could be a problem with them. The study provides a lot of information and data. Overall, a very good paper and very well written.

I really don have any major issues with the paper. Somehow the formatting of Table 2 did not allow its complete printing, but I did not find any problem with the contents. I would like to see larger fonts in the legends of Fig. 2. Also, I am not sure if you really need lines 92 to 101.It is unusual to provide these mini introductions to the other sections of the manuscripts, but it could be the writing style of the authors and I will leave it to them.

Best wishes.

6. PLOS authors have the option to publish the peer review history of their article (what does this mean?). If published, this will include your full peer review and any attached files.

Reviewer #1: No

Reviewer #2: No

---

## [Author Response · Author response to Decision Letter 0]

18 Jan 2021

Dear Dr. Zang,

We are submitting a revised version of PONE-D-20-31336. We thank you and the reviewers for 

thoughtful comments and suggestions. We’ve responded to the comments point by point below. 

Note that in our responses, the line numbers refer to the tracked changes version of the resubmitted 

manuscript.

Editor comments

1. PLOS style requirements:

Response: We’ve revised our file names to try to follow the requirements. In general, we

believe we have satisfied the style requirements. Please let us know of anything we have

overlooked.

2. We note that Figures 1-5 in your submission contain map images which may be copyrighted.

Response: We’re not sure why you think the figures may be copyrighted. We created these

figures using the data presented in this work. The copyright is held by us (the authors), so

we shouldn’t need to seek permission from anyone.

3. Please include captions for your Supporting Information files at the end of your manuscript,

and update any in-text citations to match accordingly.

Response: All of the tables in our Supporting Information files have captions (we have two

tables in S3 Appendix, an S4 Table, and no supporting figures). So we’re not sure what this

request is referring to. We’re happy to make modifications if given additional guidance.

Reviewer #1 comments

• In my opinion, the “Methods” part is too long and has many detailed information and de-

scription, it may be better to present the Methods in a concise way and move the detailed

description into a supplementary material. In addition, there are a few points the authors

need to clarify in their manuscript.

Response: We’ve moved a number of additional details about the data cleaning/preprocessing

(lines 159-187 and 210-237 in the original submission) to the supplementary material (lines

9-41, 237-265 in S1 Appendix). If required, we could move more details of the statistical

smoothing as well. However, the smoothing approach we have taken is a new statistical ap-

proach to such forest structure data and thus represents a core part of what is new about this

work compared to [7], so we feel it appropriate to leave this in the main methods section.

• 1. Line 51, give a concise description of the “Historical datasets”, what kinds of data are

included in the “Historical datasets”.

Response: We’ve changed “datasets” to be “vegetation surveys” in line 51 of the resubmis-

sion to be more precise. Combined with the extensive discussion of the survey data in lines

65-76 of the resubmission, we feel that the description of the data used is clear.

• 2. Line 57, from line 53 to 57, the authors listed several aspects in which the datasets are

potentially useful and could be applied to study those questions. I suggest the authors illus-

trate the applications of their dataset to those questions using proper examples.

Response: We’ve added a number of citations in lines 54-57 of the resubmission that illus-

trate such uses of these datasets.

• 3. Line 87, briefly describe the results obtained by [7], and explain the relationship between

this work and [7], and highlight the novelty and significance of the current work.

Response: We’ve added more detail on what was done in [7] in lines 87-91 of the resubmis-

sion. We’ve retained previous text (lines 91-95 in the resubmission) that also discussed how

these results differ from [7]. Finally, we added text (lines 95-96 of the resubmission) to indicate

that in this work we focus on analyzing the biogeographic patterns at the time of settlement

(i.e., with less focus on comparing to the modern vegetation than in [7]).

• 4. Line 101, from line 92 to 101, the authors summarized the main idea of their manuscript.

I suggest the authors clearly present the significance of their work at the end of the “Intro-

duction”.

Response: We’ve added text in lines 109-115 of the resubmission to this effect.

Reviewer #2 comments

• Somehow the formatting of Table 2 did not allow its complete printing, but I did not find any

problem with the contents.

Response: Yes, we are a bit confused about how to present these wide tables. We tried to

follow the PLOS requirements but welcome further guidance.

• I would like to see larger fonts in the legends of Fig. 2.

Response: We have increased the font size.

• Also, I am not sure if you really need lines 92 to 101. It is unusual to provide these mini

introductions to the other sections of the manuscripts, but it could be the writing style of the

authors and I will leave it to them.

Response: the first author is a statistician, and it is common in statistics papers to present an

overview of the remainder of the paper at the end of the introduction. We’d like to leave this

material, but if the Editor requests that we remove it, we will do so.

Other changes

In revising the manuscript, we have also made some other minor wording changes in various places

in the document. These can of course be seen in the tracked changes version of the main manuscript

and the S1 Appendix and S2 Appendix.

---

## [Editor Report · Decision Letter 1]

20 Jan 2021

The forests of the midwestern United States at Euro-American settlement: spatial and physical structure based on contemporaneous survey data

PONE-D-20-31336R1

Dear Dr. Paciorek,

We’re pleased to inform you that your manuscript has been judged scientifically suitable for publication and will be formally accepted for publication once it meets all outstanding technical requirements.

Kind regards,

RunGuo Zang

Academic Editor

PLOS ONE

Additional Editor Comments (optional):

accept
---

## [Editor Report · Acceptance letter]

28 Jan 2021

PONE-D-20-31336R1 

The forests of the midwestern United States at Euro-American settlement: spatial and physical structure based on contemporaneous survey data 

Dear Dr. Paciorek:

I'm pleased to inform you that your manuscript has been deemed suitable for publication in PLOS ONE. Congratulations! Your manuscript is now with our production department. 

Kind regards, 

on behalf of

Professor RunGuo Zang 

Academic Editor

PLOS ONE